# [Re]: On the Relationship between Self-Attention and Convolutional Layers

**Mukund Varma T**
Indian Institute of Technology Madras
mukundvarmat@gmail.com

**Nishant Prabhu**
Indian Institute of Technology Madras
me17b084@smail.iitm.ac.in

## Abstract

In this report, we perform a detailed study on the paper "On the Relationship between Self-Attention and Convolutional Layers", which provides theoretical and experimental evidence that self attention layers can behave like convolutional layers. The proposed method does not obtain state-of-the-art performance but rather answers an interesting question - *do self-attention layers process images in a similar manner to convolutional layers?* This has inspired many recent works which propose fully-attentional models for image recognition. We focus on experimentally validating the claims of the original paper and our inferences from the results led us to propose a new variant of the attention operation - *Hierarchical Attention*. The proposed method shows significantly improved performance with fewer parameters, hence validating our hypothesis. To facilitate further study, all the code used in our experiments are publicly available here[1].

## 1 Introduction

In computer vision, convolutional architectures [1, 2, 3, 4] have dominated across various image recognition tasks like classification, segmentation, etc. However, they have some limitations like lack of rotation invariance, inability to aggregate information based on the image content, etc. This has inspired researchers to explore a different design space and introduce models with interesting new capabilities. Self-attention based networks, in particular Transformers [5] have become the model of choice for various Natural Language Processing (NLP) tasks. The major difference between transformers and previous methods, such as recurrent neural networks and Convolutional Neural Networks (CNN), is that the former can simultaneously learn to attend to various parts of the input sequence. To utilize this capacity to learn meaningful inter-dependencies, many recent works have tried to incorporate self-attention [6, 7], some even replacing convolutions entirely [8, 9], in networks for vision tasks. The main highlights of the paper "On the Relationship between Self-Attention and Convolutional Layers" [10] are:

1. Theoretical proof that self-attention layers can behave similar to convolution layers.
2. Empirical validation of the performance of the self-attention model on the image classification task using the CIFAR10 dataset. Visual evidence that self-attention applied to images learns convolutional filters around the query pixels.

In this study, we perform a detailed analysis on the various experiments outlined in the paper. We observe certain differences from the original paper which lead to interesting observations. We go beyond verifying the claims by trying to solve the observed problems and propose a novel attention operation, which we refer to as *Hierarchical Attention* (HA). We incorporate HA in various existing architectures [10, 11, 12] and our detailed experiments suggest significantly improved performance ($\approx 5\%$) with roughly $1/5^{th}$ the number of parameters.

### 1.1 Outline of this report

We structure the report as follows:

1. Section 2 formally introduces the attention operation, followed by the fundamental principle of the transformer.
2. We validate the claims made by the paper in Section 3. We visualize the attention patterns of the suggested models and compare with those mentioned in the paper, commenting on any similarities or differences.
3. We introduce a novel Hierarchical Attention operation described in Section 4. We compare the modified operation on various methods and empirically show significant performance gains.
4. Section 5 concludes and suggests possible improvements for future research.

---

[1] https://github.com/NishantPrabhu/Self-Attention-and-Convolutions

## 2   Fundamentals

In this section, we introduce the attention operation, first in terms of its origin in NLP and how it can be extended for images. We also provide a short introduction to transformers since most methods described in this report heavily rely on it.

### 2.1   Attention

Attention was first proposed for NLP, where the goal is to focus on a subset of important words. Consequently, relations between inputs are highlighted that can be used to capture context and higher-order dependencies. The attention matrix $A(.)$ indicates a score between $N$ queries $Q$ and $N_k$ keys, which indicates which part of the input sequence to focus on. $\sigma(.)$ is an activation function (generally $softmax(.)$).

$$A(Q, K) = \sigma(QK^T) \tag{1}$$

To capture the relations among the input sequence, the values $V$ are weighted by the scores from Equation 1. Therefore, we have

$$\text{SelfAttention}(Q, K, V) = A(Q, K) \cdot V \tag{2}$$

While in NLP each element in the sequence corresponds to a word, the same idea is applicable for a sequence of N discrete objects, like pixels of an image. A key property of the self-attention model described above is that it is equivariant to the input order, i.e. it gives the same output independent of how the $N$ input tokens are shuffled. This is quite problematic in cases where the order actually matters like in images. Hence, a positional encoding is learnt for each token in the sequence and added before the self-attention operation. Hence, the $Q$ and $K$ vectors are derived from the summation of input $X$ and positional encoding $P$.

### 2.2   Transformer Attention

The Transformer network is an extension of the attention mechanism from Eqn. 2 based on the Multi-Head Attention (MHA) operation. Rather than computing the attention once, the MHA operation computes it multiple times (heads). This helps the transformer jointly attend to different information derived from each head. The output from each of these heads are concatenated before projecting onto a final output dimension. A transformer layer also contains a residual connection followed by a layer normalization. The overall operation can be summarized as:

$$\text{MHA}(Q, K, V, heads) = \text{concat}_{heads}[A(Q, K, V)]$$
$$\text{Transformer} = \text{LayerNorm}(\text{MHA} + \text{MLP}(\text{MHA})) \tag{3}$$

### 2.3   Positional Encoding for Images

There are two types of positional encodings used in transformer-based architectures: absolute and relative encoding. Absolute encodings assign a (fixed or learned) vector $P_p$ to to every pixel $p$ whereas the relative positional encoding [13] considers only the position difference between the query pixel (pixel we compute the representation of) and the key pixel (pixel we attend to).

The authors from the paper have elegantly proved how the attention operation mimics a convolution. The main result is the following: *A multi-head self-attention layer with $N_h$ heads of dimension $D_h$, output dimension $D_{out}$ and a relative positional encoding of dimension $D_p >= 3$ can express any convolutional layer of kernel size $\sqrt{N_h} * \sqrt{N_h}$ and $min(D_h, D_{out})$ output channels*. In this proposed construction, the attention scores of each head must attend to different relative pixel shifts within a kernel (Lemma 1 from the paper). The above condition is satisfied for the relative positional encoding referred to as *Quadratic Encoding*. However, experiments suggest that a relative position encoding learned by a neural network (*Learned Relative Position Encoding*) can also satisfy the conditions of the lemma. We strongly urge the readers to refer to the original paper to get a complete understanding.

## 3   Reproducibility

The aim of this section is to validate the results claimed by the paper - to examine whether self-attention layers in practice do actually learn to operate like convolutional layers when trained on the standard image classification task. For all our experiments mentioned in this report, we use the CIFAR10 dataset to benchmark the performance of the model. We implement the original paper from scratch in PyTorch and refer to the author's source code for verification[2].

---

[2]`https://github.com/epfml/attention-cnn`

### 3.1 Dataset - CIFAR10

The CIFAR-10 dataset [14] consists of 60,000 colour images of size 32x32 split in 10 classes. There are 6,000 images per class split into 5,000 training and 1,000 validation samples.

### 3.2 Computational Requirements

Experiments involving smaller models, namely ResNet18 and Quadratic embedding, were trained on an 8GB NVIDIA RTX 2060 GPU. Learned embedding-based models were trained on 16GB NVIDIA V100 virtual GPUs rented from Amazon Web Services (AWS). Each training run for the smaller models required around 20 hours while the larger experiments took over 2 days for convergence.

### 3.3 Experiments and Results

The results mentioned in the paper uses a fully-attentional model consisting of 6 multi-head self-attention layers each with 9 heads. In all the experiments, the input image undergoes a 2x2 down-sampling operation to reduce its size. The final image vector is derived by average-pooling the representations derived from the last layer and then passed to a linear layer for classification. Please refer to Table. 4 (Appendix) for a detailed list of hyperparameters used in each experiment. We closely refer to the official implementation and were able to reproduce all the results within 1% of the reported value. Table. 1 compares the results mentioned in the paper and the ones obtained using our implementation. Fig. 1a visualizes the test accuracy on CIFAR10 at every 10 epochs for each model and it is quite evident that fully convolutional networks like ResNet18 tend to converge faster. The following subsections describe these results in detail.

| Model | Accuracy (paper) | **Accuracy (ours)** | # of params |
|---|---|---|---|
| ResNet18 | 0.938 | **0.946** | 11.2M |
| Quadratic embedding (isotropic) | 0.938 | **0.943** | 12.1M |
| Quadratic embedding (non-isotropic) | 0.934 | **0.939** | 12.1M |
| Learned embedding w/o content | 0.918 | **0.904** | 12.3M |
| Learned embedding w/ content | 0.871 | **0.864** | 29.5M |

Table 1: Test accuracy (paper vs ours) on CIFAR-10 and model sizes; 9 heads

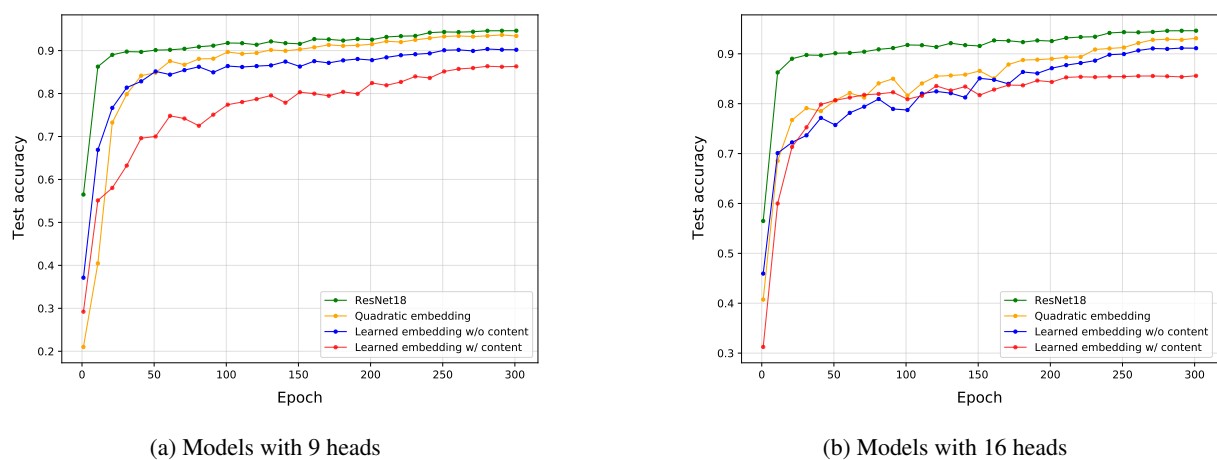

(a) Models with 9 heads        (b) Models with 16 heads

Figure 1: Test performance on CIFAR10 at every 10 epochs

### 3.3.1 Quadratic Encoding

The authors show that the attention probabilities in the quadratic positional encoding is similar to an isotropic bivariate Gaussian distribution with bounded support. Hence to validate their claims, all the attention matrices in the model are replaced with these Gaussian priors, with learnable parameters to determine the center and width of each attention head. Further, this is extended to a non-isotropic distribution over pixel positions as it might be interesting to see if the model would learn to attend to such groups of pixels - thus forming unseen representations in CNNs. Fig. 2 visualizes the attention centers for each head for all the layers and at different epochs. After optimization, we can see that the heads attend to a specific pixel of the image forming a grid around the query pixel. This confirms the intuition that self-attention applied to images learns convolution-like filters around the query pixel. Also, it can be seen that the initial layers (1-2) focus at local patterns while the deeper layers (3-6) attend to larger patterns by positioning the center of attention further from the queried pixel position. Fig. 2b shows that the network did learn non-isotropic attention patterns especially in the last layers. However, there is no performance improvement suggesting that it is not particularly helpful in practice.

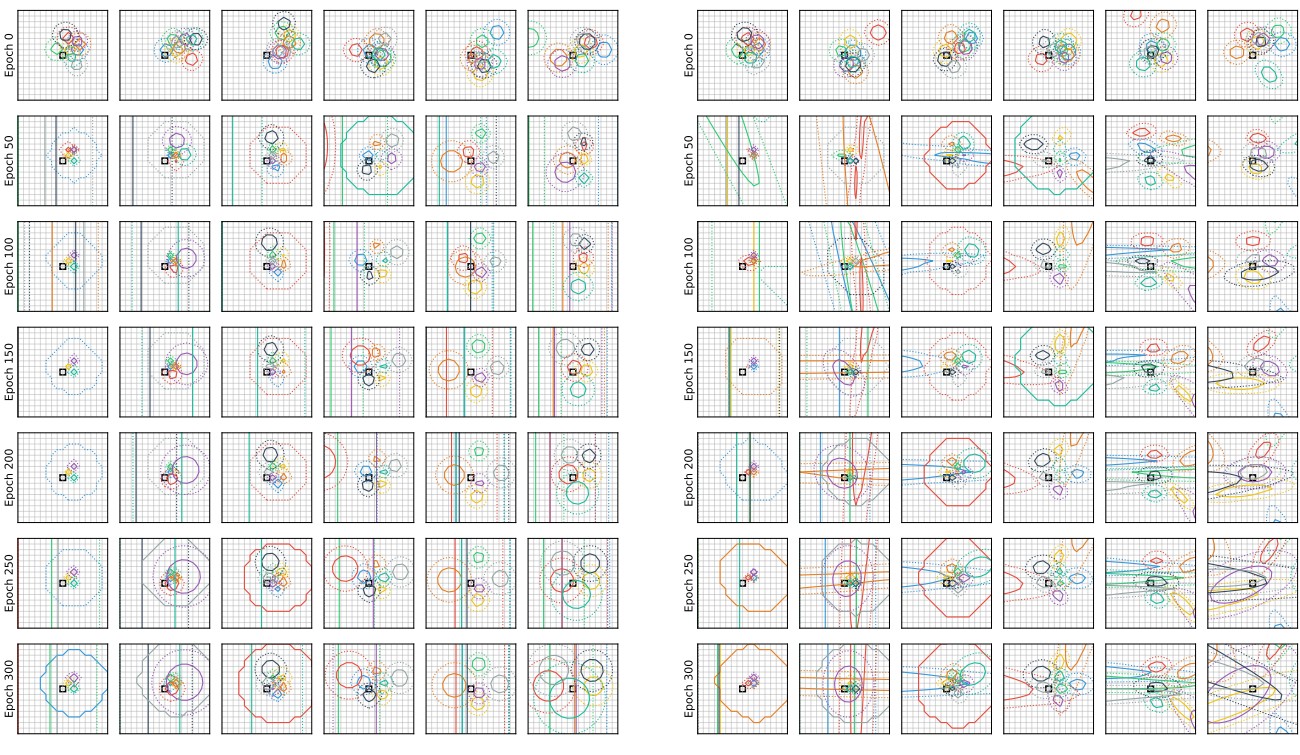

(a) isotropic Gaussian parameterization        (b) non-isotropic Gaussian parameterization

Figure 2: Centers of attention of each attention head (different colors) for all 6 layers (*columns*) at various training epochs (*rows*). The central black square is the query pixel, whereas solid and dotted circles represent the 50% and 90% percentiles of each Gaussian, respectively.

.

### 3.3.2 Learned Relative Position Encoding

In this experiment, the authors try to study the positional encoding generally used in fully-attentional models [8]. The positional encoding vector for each row and column pixel shift is learnt. The final relative position encoding of a key pixel with a query pixel is derived as the concatenation of row and column shift embeddings. First, the authors completely discard the input data and compute the attention weights solely with the derived encoding (Learned embedding w/o content). Fig. 3a visualizes the attention probabilities for a given query pixel, confirming the hypothesis that: even when left to learn positional encoding from randomly initialized vectors, certain self-attention heads learn to attend to individual pixels while the others learn non-localized patterns and long range dependencies. In another setting (Learned embedding w/ content), both the positional and content based attention information is used which corresponds to a full-blown stand alone self-attention model. Fig. 3b visualizes the attention probabilities for a given query pixel in this setting and it is interesting to note that even when left to learn the encoding from the data, some attention heads

exploit positional information like CNNs while the others focus on the content. In Fig. 4, we visualize the attention probabilities averaged across an entire batch of images to understand the focus of each head and remove dependency on the input image for both the experiments.

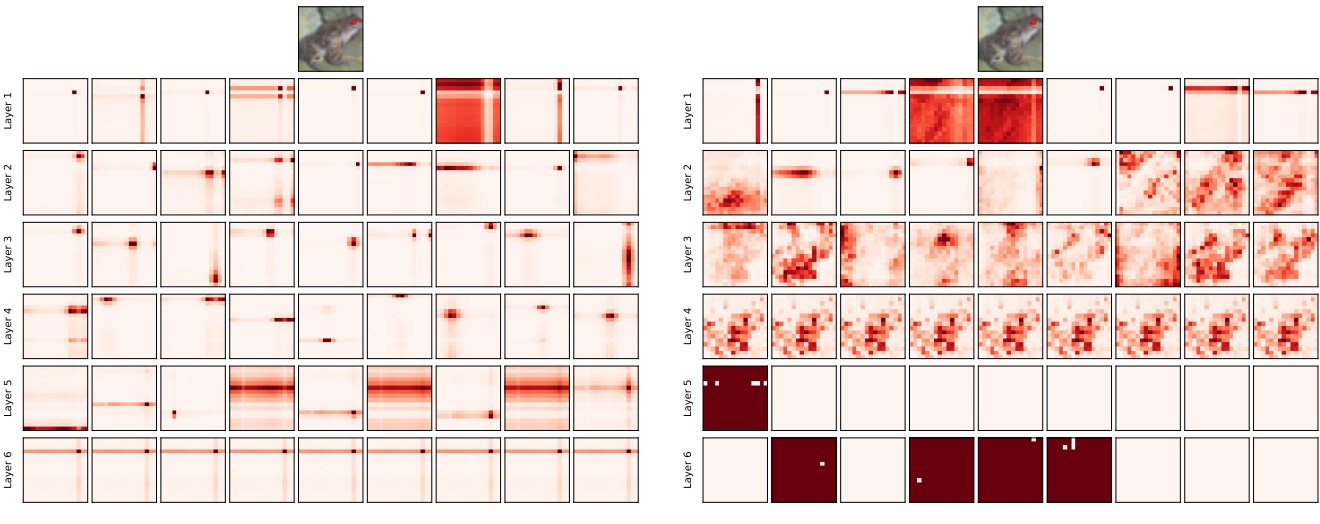

(a) Learned embedding w/o content          (b) Learned embedding w/ content

Figure 3: Attention probabilities for a model with 6 layers (rows) and 9 heads (columns) using learned relative positional encoding (with and without content). The query pixel (red square) is on the frog head.

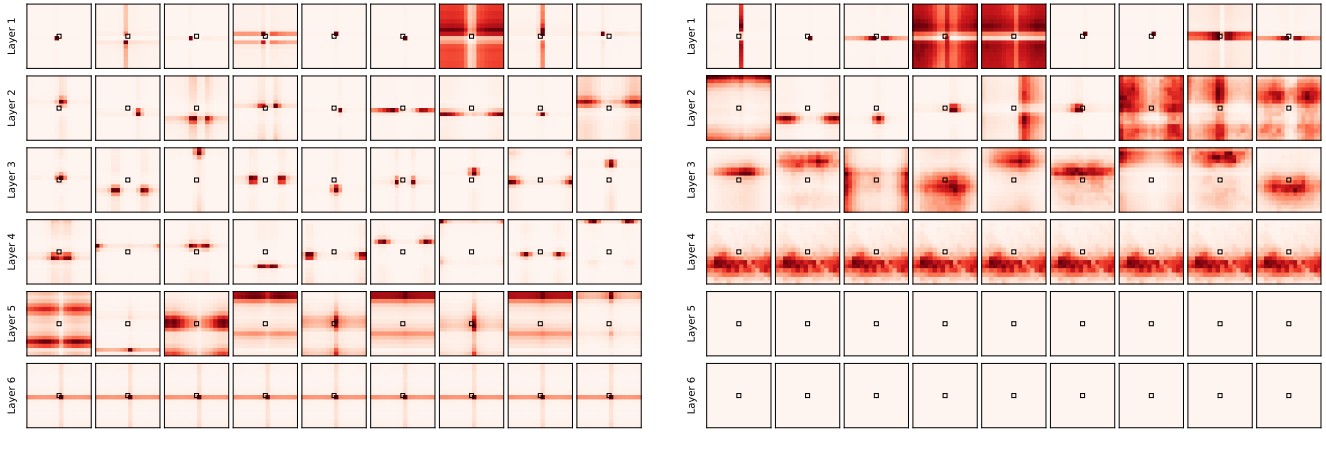

(a) Learned embedding w/o content          (b) Learned embedding w/ content

Figure 4: Attention probabilities for a model with 6 layers (rows) and 9 heads (columns) using learned relative positional encoding (with and without content). Attention maps are averaged over 50 test images to display head behavior and remove the dependence on the input content. The query pixel is in the center of the image.

**Average Attention Visualization**: The authors from the original paper visualize the attention probabilities for a single image or across a batch images for a specific query pixel. A single query pixel does not convey information regarding where the model focuses on in the entire image and it is not practical to plot individual figures for every query pixel. Hence, we also visualize the attention probabilities using what we refer to as *Average Attention*, to identify which portions of the entire image the model attends to. Given a softmax normalized attention matrix of size $N \times N$, every row represents the relationships between a query pixel and the others. First, every element ($\alpha_{i,j}$) is divided by the row-wise sum to ensure that all the values are in scale. Then the row-wise mean is computed to determine the importance value for each pixel. If a pixel is strongly correlated to multiple pixels, the importance value will be higher determining that the model has a stronger focus on that given pixel. We mathematically describe the operation in Eqn. 4. Fig. 5 visualizes the average attention for the learned embedding with and without content. In Fig. 5a, since the content data is

discarded, the model is clearly focusing on positional patterns while in Fig. 5b, the model attends to both positional and content information. We visualize additional figures in Sections. B.1-B.3 (Appendix).

$$\alpha_{i,j} = \text{softmax}(\alpha_{i,j}) = \frac{\exp \alpha_{i,j}}{\sum_k (\alpha_{i,k})}$$
$$\alpha_{i,j} = \frac{\alpha_{i,j}}{\sum_k (\alpha_{k,i})} \tag{4}$$
$$\text{Avg Attn}_i = \frac{\sum_k (\alpha_{i,k})}{N}$$

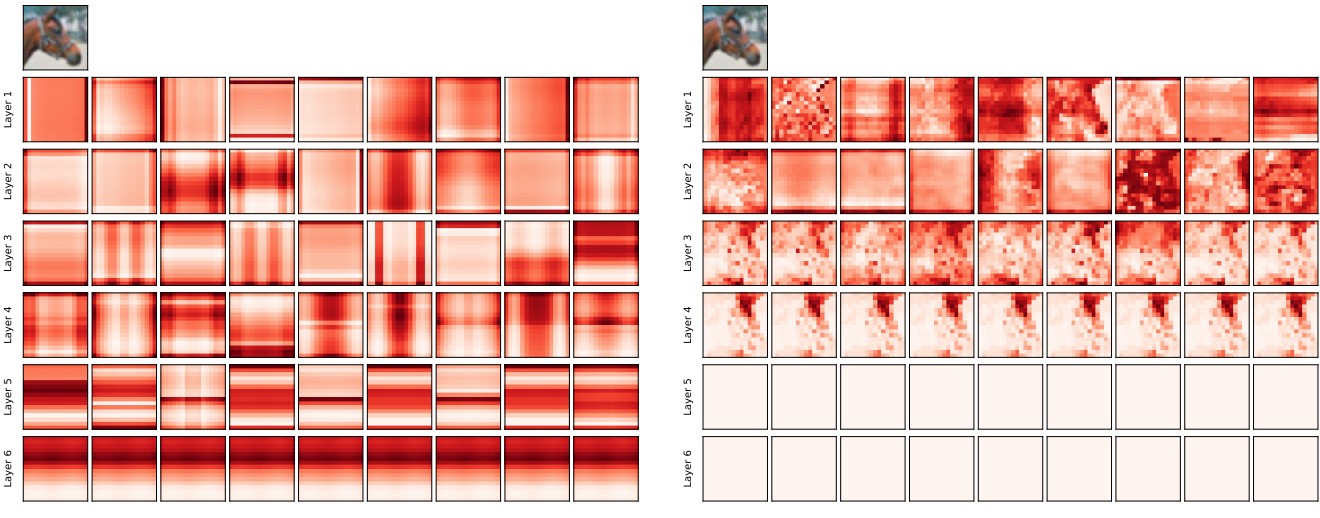

(a) Learned embedding w/o content          (b) Learned embedding w/ content

Figure 5: Average Attention visualization for a model with 6 layers (rows) and 9 heads (columns) using learned relative positional encoding (with and without content).

### 3.4 Increasing the number of heads

As per the analogy derived between self-attention and convolutions, the number of heads is directly related to the kernel size in a convolution operation. Hence, we increase the number of heads from 9 to 16. It is important to note here that the unlike the general procedure of setting $D_h = D_{out}/N_h$ in transformer-based architectures, the paper suggests to concatenate heads of dimension $D_h = D_{out}$ since the effective number of learned filters is $min(D_h, D_{out})$. Given the limited compute, we reduced $D_{out}$ from 400 to 256 while increasing the number of heads to 16. As seen in Table. 2 there seems to be no significant impact to the model's performance. However, the model takes longer to converge due to the increased number of parameters (Fig. 1b). We visualize the attention probabilities in Sections. B.4-B.6 (Appendix).

| Model | Accuracy |
|---|---|
| Quadratic embedding | 0.931 |
| Learned embedding w/o content | 0.912 |
| Learned embedding w/ content | 0.856 |

Table 2: Test accuracy on CIFAR-10; 16 heads

### 3.5 Additional Observations

#### 3.5.1 Inductive biases in Transformers

As seen from the results mentioned in Table. 1, the fully attentional model utilizing learnable embeddings with image content performs poorly when compared to the other methods. As mentioned in the paper [11], transformers lack biases inherent to CNNs like translation equivariance and retention of 2D neighborhood structure, etc. Only the Multi-Layer Perceptron (MLP) layers used in these methods are local and translation equivariant, while the self attention layers are global. Even in the case of NLP, almost 75-90% of the predictions are correct even when the input words are randomly shuffled [15]. This implies that transformer-based methods do not sufficiently capture spatial information even with positional encodings and require a large amount of training data to do so. This could be the reason for improved performance in the case of Learned embedding w/o content and Quadratic embedding as the attention matrices are directly replaced with positional information.

#### 3.5.2 Over-expressive power of attention matrices

The most important step of the self-attention operation is the generation of the attention matrix of size $N \times N$. In NLP, the value of $N$ tends to be small (<100) in most cases as we are dealing with words in a sentence. On the contrary, images when broken down result in very long sequences of pixels, hence creating large attention matrices. Therefore, these attention matrices can be sparse and the model has a very high tendency of focusing on very high level information. This can lead to over-fitting and is talked about in the case of pointclouds[3] where the number of points are very large (>1000). This has also been observed in our experiments. As seen in Figs. 3b, 4b, 5b, the attention heads in the last few 2 layers are very sparse and do not capture any information. This can also be seen in the case of Quadratic encoding (Fig. 2), where certain attention heads focus on "non-intuitive" portions of the image (i.e) a thin strip of pixels or attend uniformly across a large patch of pixels. A simple and naive way to overcome this problem is to reduce the number of heads or layers but this is not an effective method as the model loses its capacity to learn strong features.

## 4 Hierarchical Attention

Given the problems described above, we need to propose a method which can avoid the over-expressive nature of independent attention heads while still being able to learn and derive strong features from the input image across layers. We now introduce a novel attention operation which we refer to as *Hierarchical Attention* (HA) operation. In the following sections, we explain the core idea behind the operation and perform detailed experiments to showcase its effectiveness.

### 4.1 Methodology

In most of the transformer-based methods, independent self-attention layers are stacked sequentially and the output from one is passed onto the next to derive the $Q$, $K$ and $V$ vectors. This allows each attention head to freely attend to specific features and derive a better representation. Deviating from these methods, the HA operation updates only the $Q$, $V$ vectors while the $K$ remains the same after each attention block. Further, the weights are shared across these attention layers inducing the transformer model to iteratively refine its representation across layers. This can be considered analogous to an unrolled recurrent neural network (RNN) as we are trying to sequentially improve the representation across layers based on the previous hidden state. This helps the model hierarchically learn complex features by focusing on corresponding portions of the $K$ vector, and aggregating required information from the $V$ vector. Figs. 6, 7 visualize the normal attention and the hierarchical attention operations respectively. For the sake of simplicity, we do not visualize the entire inner workings of the transformer like residual connections, layer normalization, etc. This is a very simple yet effective method which can be easily adapted to any existing attention-based network as described in the next section.

As mentioned earlier, there has been a lot of recent hype in replacing convolutions with attention layers. Hence, we choose two other popular and similar papers - "Exploring Self-attention for Image Recognition" [12], "An Image is worth 16x16 words: Transformers for Image Recognition at scale" [11] and apply the proposed HA operation to justify its effectiveness. For better understanding, we briefly introduce these papers in the following subsections.

---

[3] https://github.com/juho-lee/set_transformer/issues/3#issuecomment-586711062

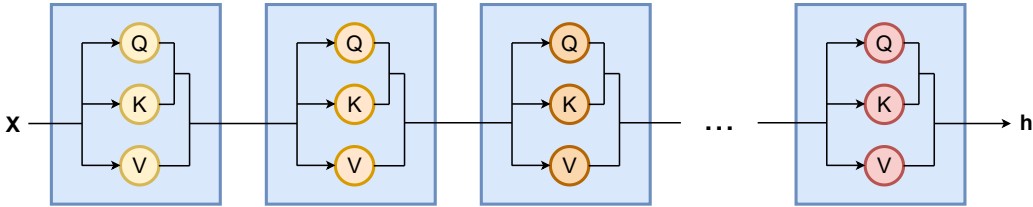

Figure 6: Normal Attention (Scaled Dot Product): The projection weights in each layer are different and independently learnt (different color). $Q, K, V$ vectors are updated in each layer.

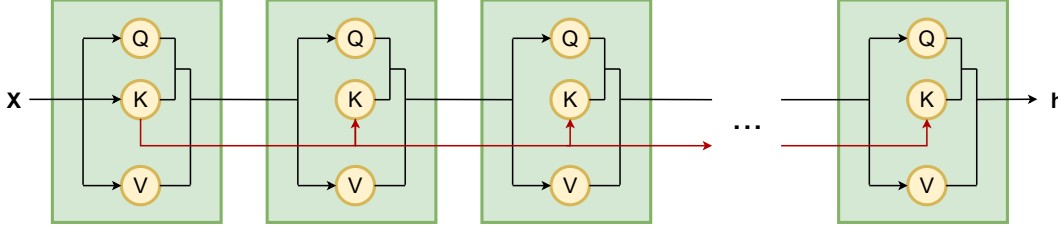

Figure 7: Hierarchical Attention (HA): The projection weights are shared across all the layers. Only Q, V vectors are updated while the K remain the same in each layer.

## 4.2 Pairwise and Patchwise self attention (SAN)

Introduced by [12], pairwise self-attention is essentially a general representation of Eqn. 2. It is fundamentally a set operation, does not attach stationary weights to specific locations and is invariant to permutation and cardinality. The paper presents a number of variants of the pairwise attention that have greater expressive power than dot-product attention. Specifically, the weight computation does not collapse the channel dimension and allows the feature aggregation to adapt to each channel. It can be mathematically formulated as follows:

$$y_i = \sum_{j \epsilon R_i} \alpha(x_i, x_j) \odot \beta\left(x_j\right)$$
$$\alpha(x_i, x_j) = \gamma(\delta(x_i, x_j)) \tag{5}$$

Here, $i$ is the spatial index of feature vector $x_i$, $\delta(.)$ is the relation function mapped onto another vector by $\gamma(.)$. The adaptive weight vectors $\alpha(x_i, x_j)$ aggregate feature vectors obtained from $\beta(.)$. An important point to note here is that $\delta$ can produce vectors of different dimensions when compared to $\beta$, allowing for a more expressive weight construction.

The patch-wise self attention is a variant of the pairwise operation, where $x_i$ is replaced by a patch of feature vectors $x_{R(i)}$ which allows the weight vector to incorporate information from all the feature vectors in the patch. The equations are hence rewritten as:

$$y_i = \sum_{j \epsilon R_i} \alpha(x_{R(i)})_j \odot \beta\left(x_j\right)$$
$$\alpha(x_{R(i)}) = \gamma(\delta(x_{R(i)})) \tag{6}$$

## 4.3 Vision Transformer (VIT)

The Vision Transformer [11], has successfully shown that reliance on convolutions is no longer necessary and a pure transformer outperforms all convolution based techniques significantly when pre-trained on large amounts of data. In this method, an image is split into patches which is then projected onto another representation using a trainable layer and then passed through a standard set of transformer operations as described in Eqn. 3.

## 4.4 Results

To validate our intuition and to prove the effectiveness of the proposed method, we incorporate HA in all the methods described above without modifying the overall structure of the architecture. In our experiments involving SAN, we utilize the official implementation[4] due to the available faster CUDA kernels while we implement VIT from scratch referring to the author's source code[5]. Table 3 compares accuracy for each model with its corresponding HA variant. We see significant improvements in the performance (at least 5% gain) in each case while being able to reduce the number of parameters to almost $1/5^{th}$ of the original model. As mentioned earlier, transformers require sufficient training to perform equally well as convolution-based architectures. When pretrained on large datasets (14M-300M images), transformer-based architectures achieve excellent performance and transfer to tasks with fewer datapoints [11]. However, for all our experiments in this report we only focus on training these models from scratch on the CIFAR10 dataset.

| Model | Normal SA | | | Hierarchical SA | | |
|---|---|---|---|---|---|---|
| | Accuracy | # of params | Wall time | Accuracy | # of params | Wall time |
| Learned embedding w/ content | 0.871 | 29.5M | 5.374 | **0.910** | 4.75M | 4.703 |
| SAN Pairwise (subtraction) | 0.818 | 3.74M | 16.586 | **0.853** | 0.35M | 12.913 |
| SAN Patchwise (subtraction) | 0.857 | 3.74M | 9.773 | **0.912** | 0.36M | 7.419 |
| Vision Transformer | 0.843 | 37.4M | 7.094 | **0.877** | 6.53M | 6.302 |

Table 3: Comparison between models using normal SA and HA. Wall times are average inference times in milliseconds for the models over 300 iterations

In Fig. 8a, we visualize the attention probabilities for a given query pixel. The relationship between self-attention and convolutions is striking as the model is attending to distinct pixels at a fixed shift from the query pixel reproducing the receptive field of the convolution operation. The initial layers attend to local patterns while the deeper layers focus on larger patterns positioned further away from the query pixel. Similarly, in Fig. 8b, the attention heads from the last two layers are no longer sparse and help capture more information. Hence the visual correctness verifies the operation and its increased performance. We also visualize the attention probabilities for SAN, VIT in Sections. B.7-B.8, B.9-B.10 (Appendix) respectively.

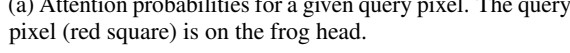
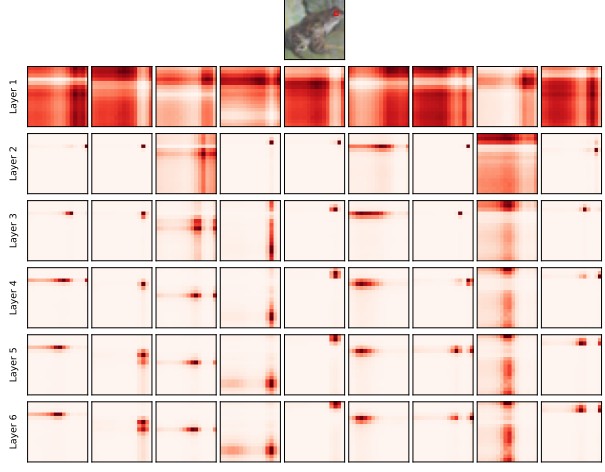

(a) Attention probabilities for a given query pixel. The query pixel (red square) is on the frog head.

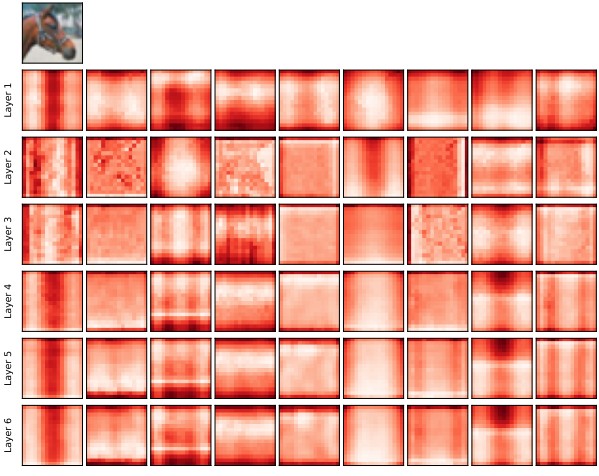

(b) Average Attention visualization

Figure 8: Attention probabilities for a model with 6 layers (rows) and 9 heads (columns) using Hierarchical Learned 2D embedding w/ content

---

[4] https://github.com/hszhao/SAN
[5] https://github.com/google-research/vision_transformer

We summarize the hierarchical operation as follows:

1. Enable weight sharing between the layers of the model by reusing the K vector and updating the Q, V vectors only. This helps the model progressively extract higher level features.

2. The method of progressive refinement ensures that the attention matrices do not get over expressive. This leads to a significant improvement over the corresponding non-hierarchical cases.

3. The total number of parameters in the model is independent of the number of layers and this property helps significantly reduce the number of parameters when compared to the non-hierarchical versions. Also, this helps make deeper models without worrying about memory constraints.

4. Even if the model is provided more layers than are necessary, every layer learns to attend to a different pattern. The new features learnt at every layer add on to those learnt by the previous ones, which provides its characteristic hierarchical nature. By visualizing the attention scores on a test image, we obtain convincing evidence to support this hypothesis.

# 5 Conclusion

In this report, we study the application of self-attention for image recognition, specifically image classification. We validate the original paper's claims by performing detailed experiments on the CIFAR10 dataset. We were able to reproduce all the results from the paper within 1% of the reported value, hence validating the claims of the original paper. However, there seems to be some differences in the attention figures which lead to interesting insights and the proposed Hierarchical Attention. To validate our hypothesis, we perform detailed experiments by incorporating HA with various methods which helps significantly improve the performance while reducing the number of parameters. These preliminary results raise various questions: Do we actually need multiple independent layers in large transformers? Does this improved performance also translate to large datasets and across various other image recognition tasks like object detection and image segmentation? We would like to answer all these questions and provide a more rigorous understanding of the proposed method in the future.

**What was easy** The original paper is well written, quite complete and mentioned the required details to carry out various experiments mentioned in the paper.

**What was difficult** The authors have possibly ported code from the transformers repository by HuggingFace[6] and left behind lots of unnecessary code making it difficult to follow. We observed major differences in the attention plots from our implementation and those reported in the paper. We attempted to contact the authors to discuss these findings, but we received no response from them. However, we provide reasoning that supports our results, which also led us to conceptualize Hierarchical Attention. For computational reasons, our scope was limited to smaller datasets like CIFAR10 with limited hyper-parameter study.

---

[6]`https://github.com/huggingface/transformers`

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

# Appendix

## A Hyperparameters

| Model | Epochs | Mini batch size | Optimizer | | Scheduler | |
|---|---|---|---|---|---|---|
| | | | Type | LR | Type | Warm up/step epochs |
| ResNet18 | 300 | 100 | SGD | 0.1 | Cosine | 10 |
| Quadratic emb. | 300 | 100 | SGD | 0.1 | Cosine | 15 |
| Learned emb. w/o content | 300 | 100 | SGD | 0.1 | Cosine | 15 |
| Learned emb. w/ content (SA/HA) | 300 | 100 | SGD | 0.1 | Cosine | 15 |
| SAN Pairwise (SA) | 300 | 256 | SGD | 0.1 | Multistep | 100, 200, 250 |
| SAN Patchwise (SA) | 300 | 256 | SGD | 0.1 | Multistep | 100, 200, 250 |
| Vision Transformer (SA) | 100 | 32 | SGD | 0.01 | Cosine | 5 |
| SAN Pairwise (HA) | 300 | 40 | SGD | 0.1 | Cosine | 10 |
| SAN Patchwise (HA) | 300 | 100 | SGD | 0.1 | Cosine | 10 |
| Vision Transformer (HA) | 100 | 32 | SGD | 0.01 | Cosine | 5 |

Table 4: Hyperparameter configuration for all experiments. SA refers to normal self-attention and HA refers to Hierarchical Attention. The momentum and weight decay for SGD were set to 0.9 and 0.0001 respectively for all experiments.

# B    Attention Visualization

We present more examples for visualising the attention probabilities in various models.

## B.1    Learned embedding w/ content; 9 heads

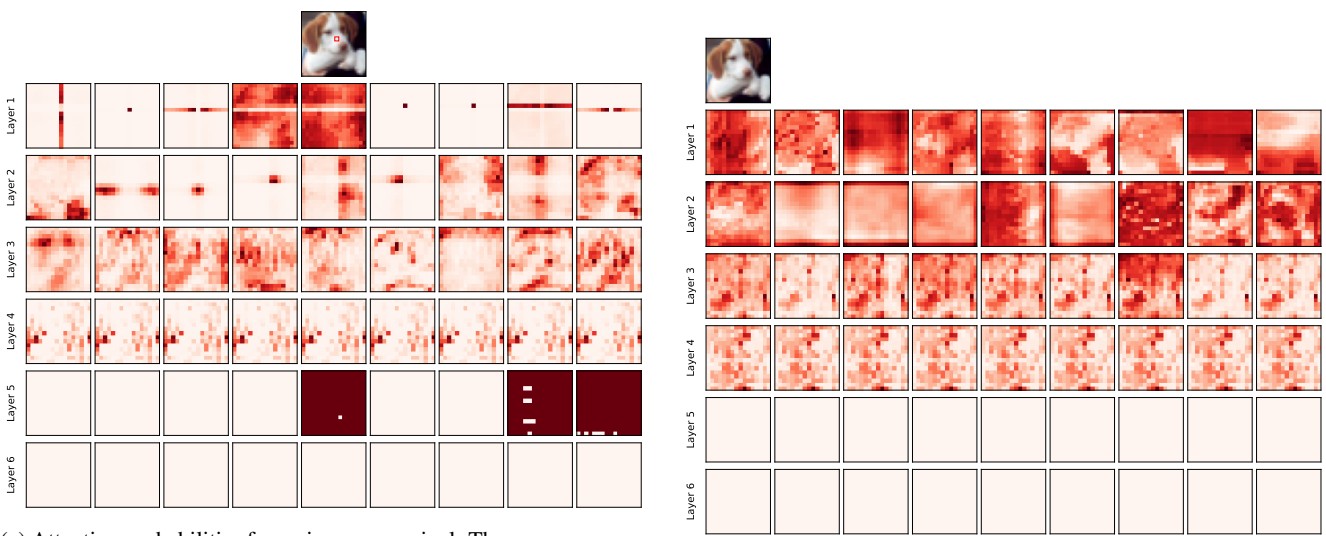

(a) Attention probabilities for a given query pixel. The query pixel (red square) is on the dog head.

(b) Average Attention visualization

Figure 9: Attention probabilities for a model with 6 layers (rows) and 9 heads (columns) using learned embedding w/ content.

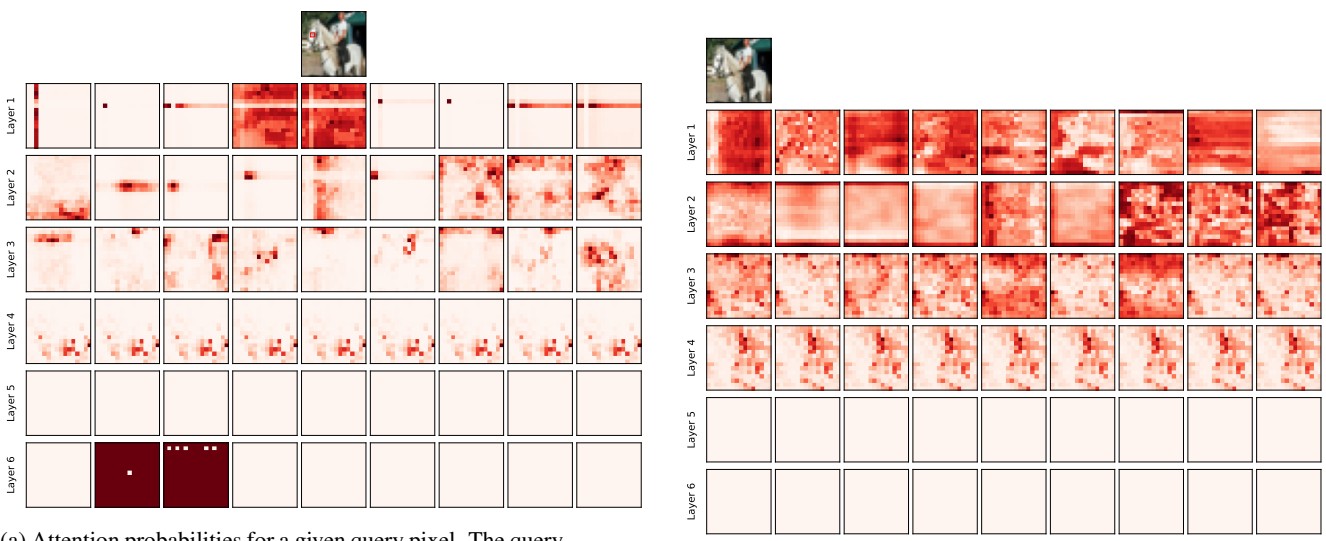

(a) Attention probabilities for a given query pixel. The query pixel (red square) is on the horse head.

(b) Average Attention visualization

Figure 10: Attention probabilities for a model with 6 layers (rows) and 9 heads (columns) using learned embedding w/ content.

## B.2 Learned embedding w/o content; 9 heads

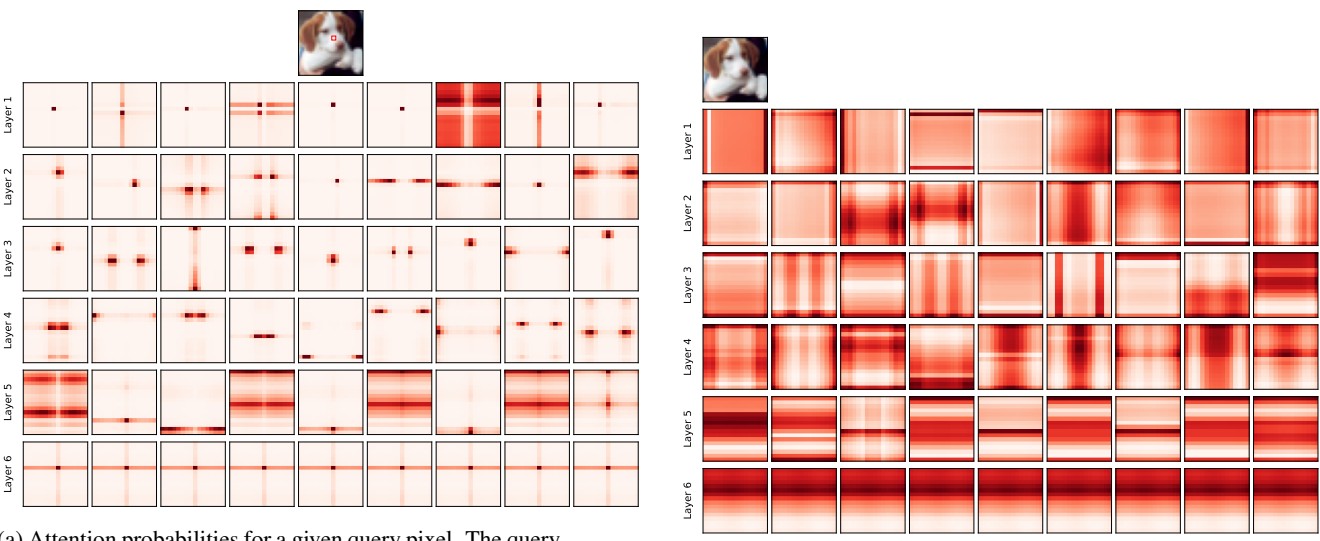

(a) Attention probabilities for a given query pixel. The query pixel (red square) is on the dog head.

(b) Average Attention visualization

Figure 11: Attention probabilities for a model with 6 layers (rows) and 9 heads (columns) using learned embedding w/o content.

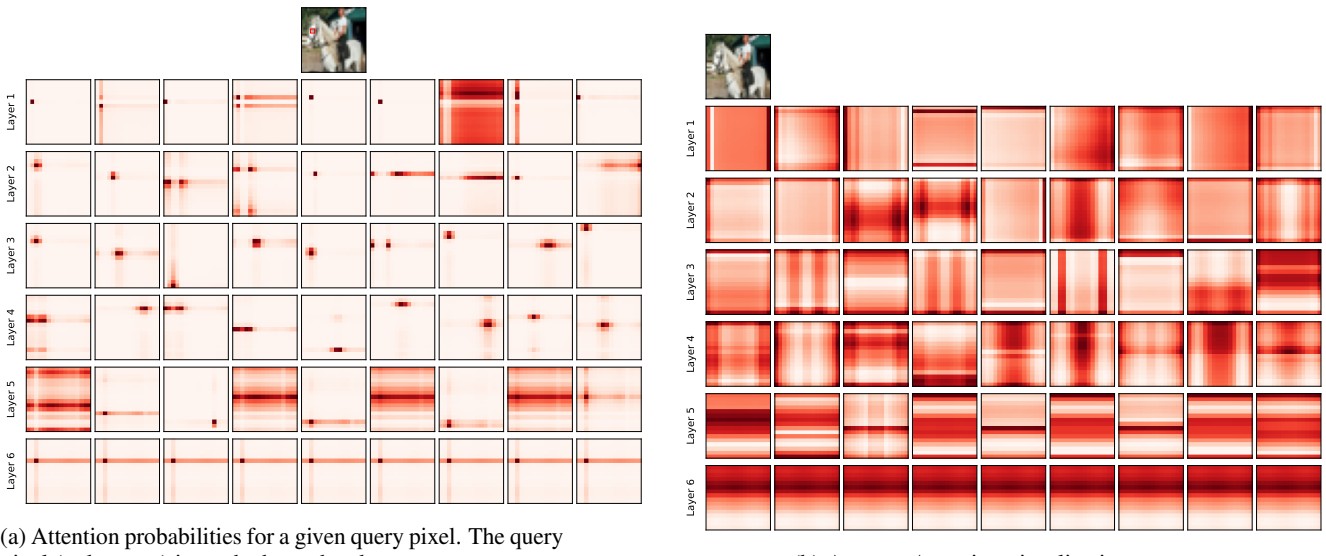

(a) Attention probabilities for a given query pixel. The query pixel (red square) is on the horse head.

(b) Average Attention visualization

Figure 12: Attention probabilities for a model with 6 layers (rows) and 9 heads (columns) using learned embedding w/o content.

## B.3 Hierarchical Learned embedding w/ content; 9 heads

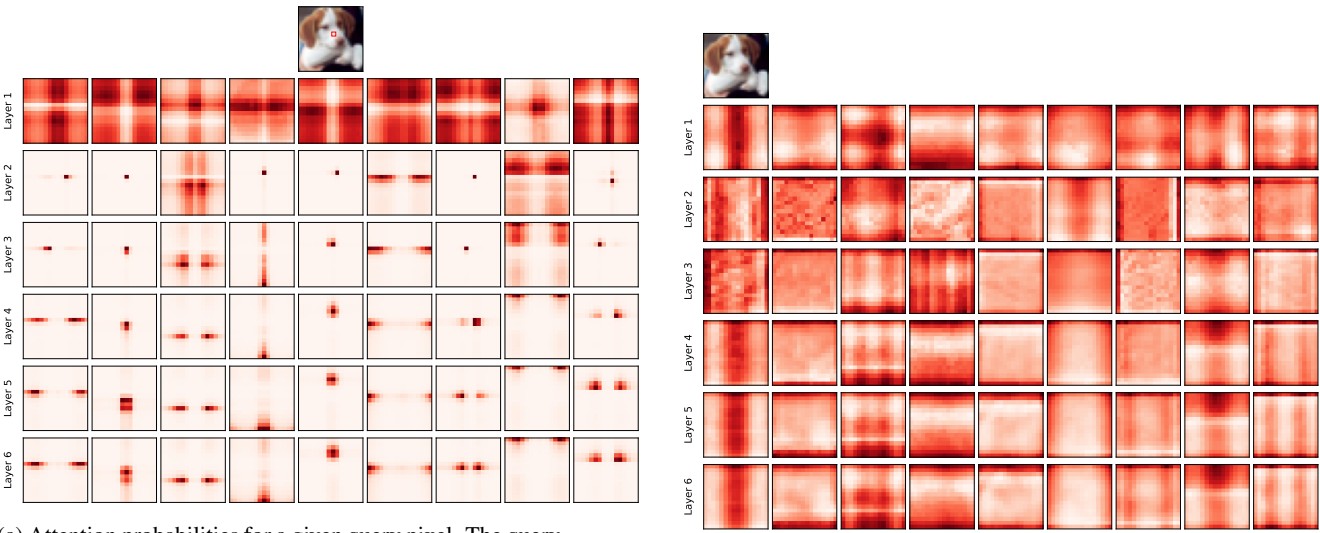

(a) Attention probabilities for a given query pixel. The query pixel (red square) is on the dog head.

(b) Average Attention visualization

Figure 13: Attention probabilities for a model with 6 layers (rows) and 9 heads (columns) using hierarchical learned embedding w/ content.

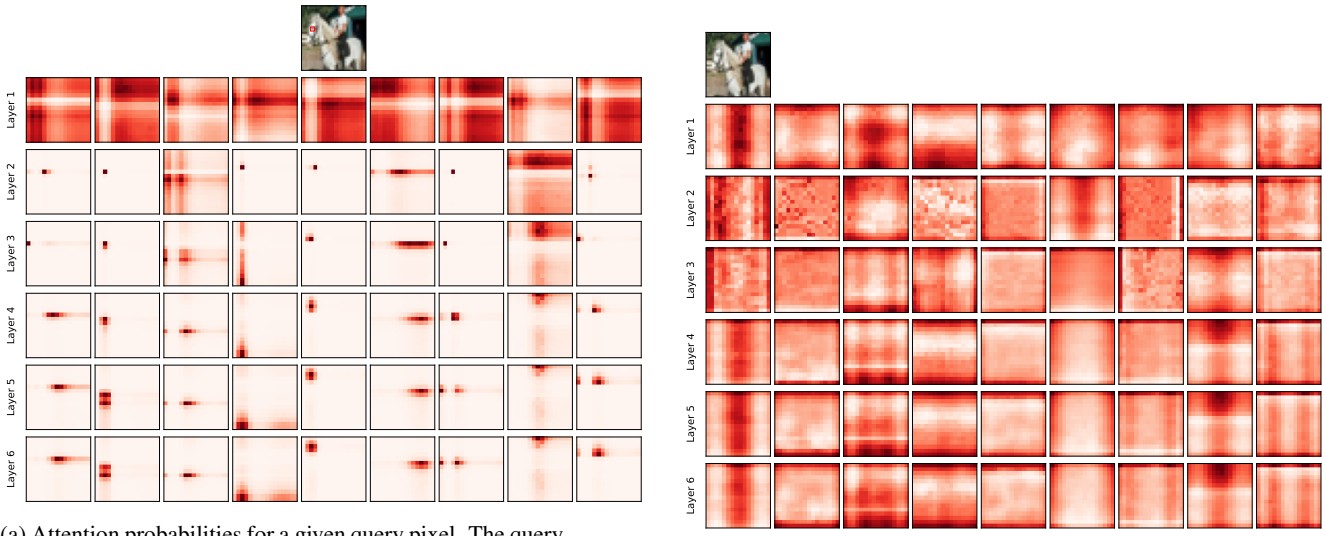

(a) Attention probabilities for a given query pixel. The query pixel (red square) is on the horse head.

(b) Average Attention visualization

Figure 14: Attention probabilities for a model with 6 layers (rows) and 9 heads (columns) using hierarchical learned embedding w/ content.

## B.4 Learned embedding w/ content; 16 heads

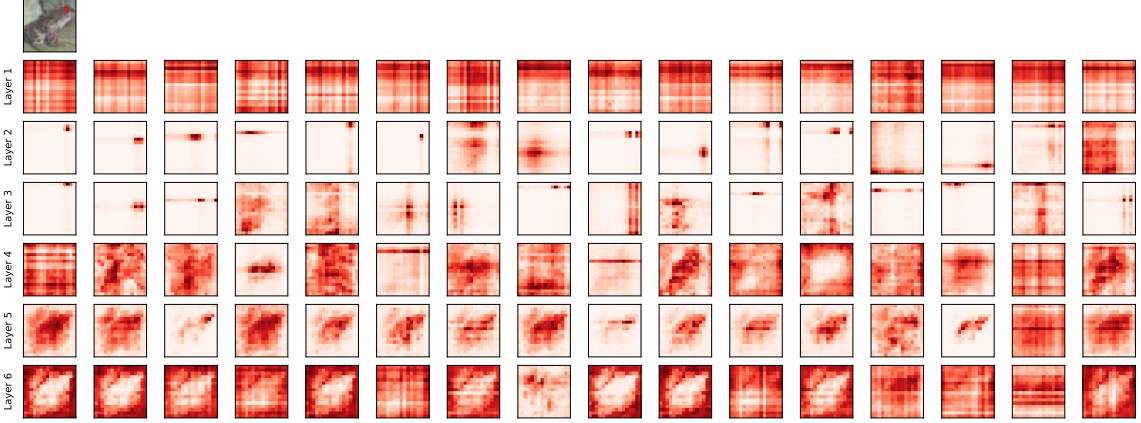

(a) Attention probabilities for a given query pixel. The query pixel (red square) is on the frog head.

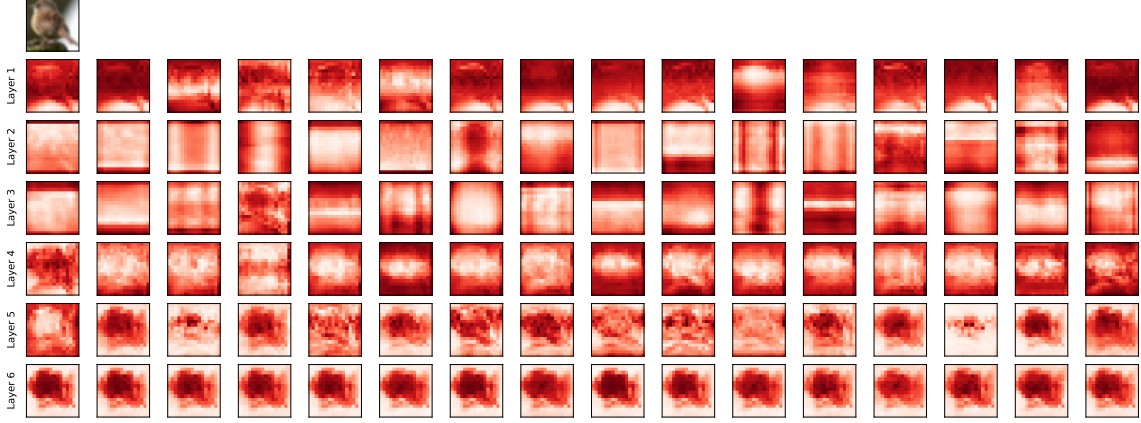

(b) Average Attention visualization

Figure 15: Attention probabilities for a model with 6 layers (rows) and 16 heads (columns) using learned embedding w/ content.

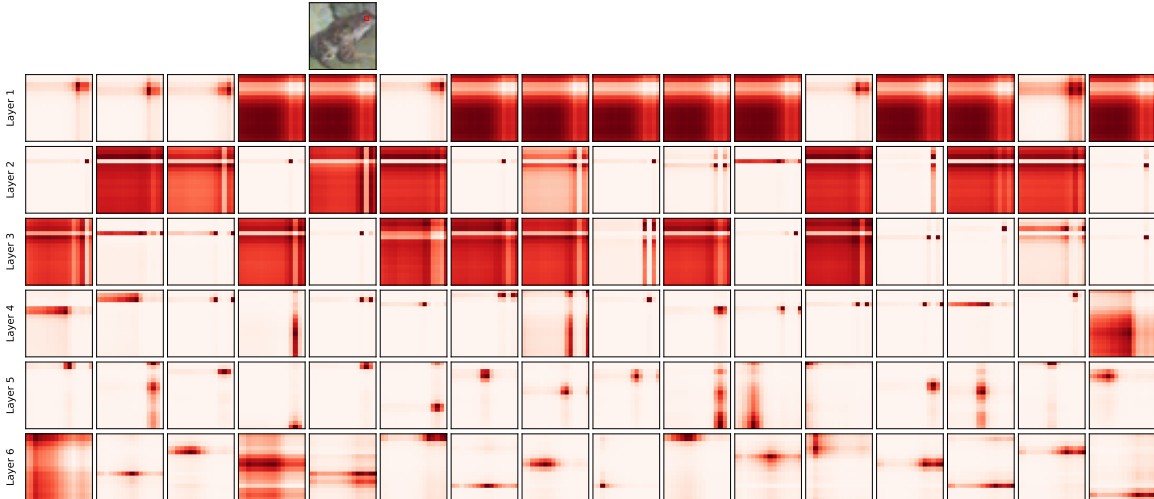

(a) Attention probabilities for a given query pixel. The query pixel (red square) is on the frog head.

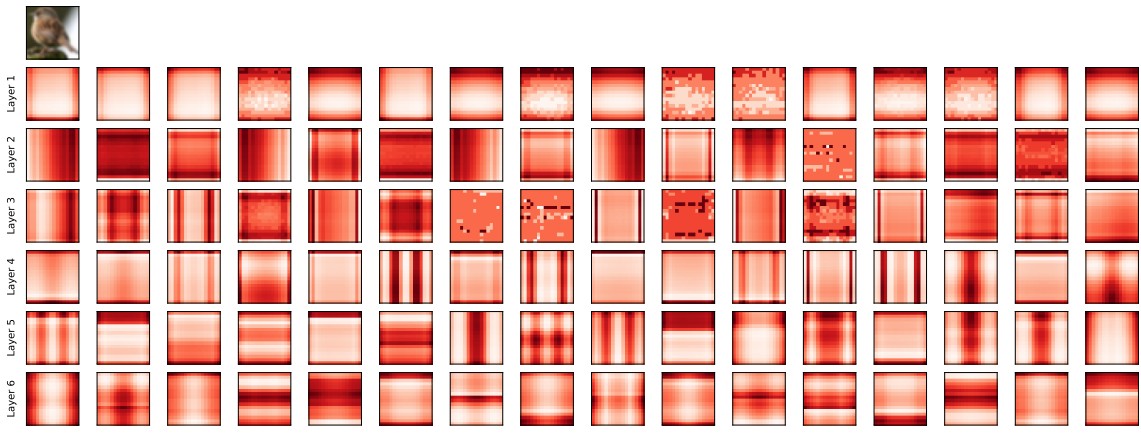

(b) Average Attention visualization

Figure 16: Attention probabilities for a model with 6 layers (rows) and 16 heads (columns) using learned embedding w/o content.

## B.6 Hierarchical Learned embedding w/ content; 16 heads

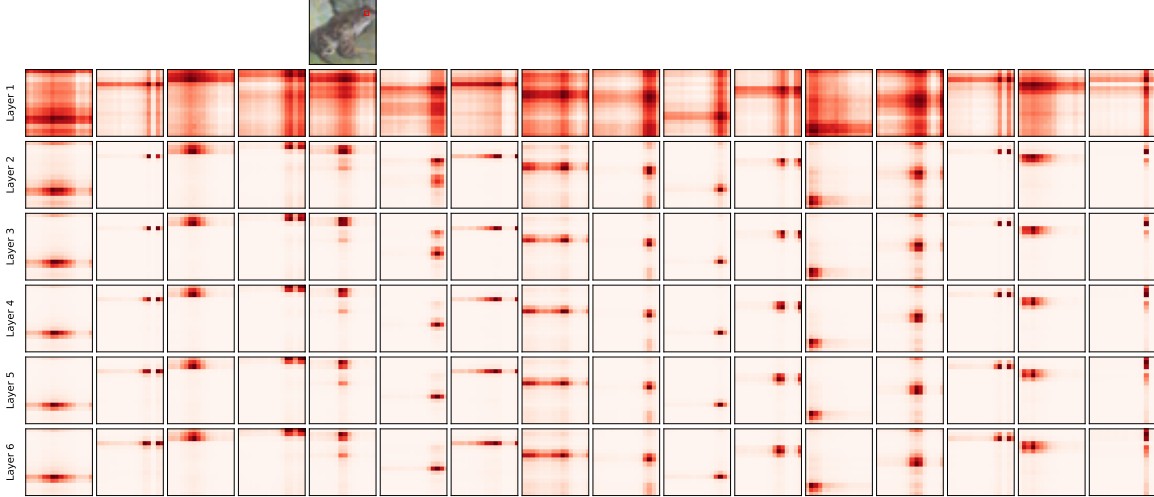

(a) Attention probabilities for a given query pixel. The query pixel (red square) is on the frog head.

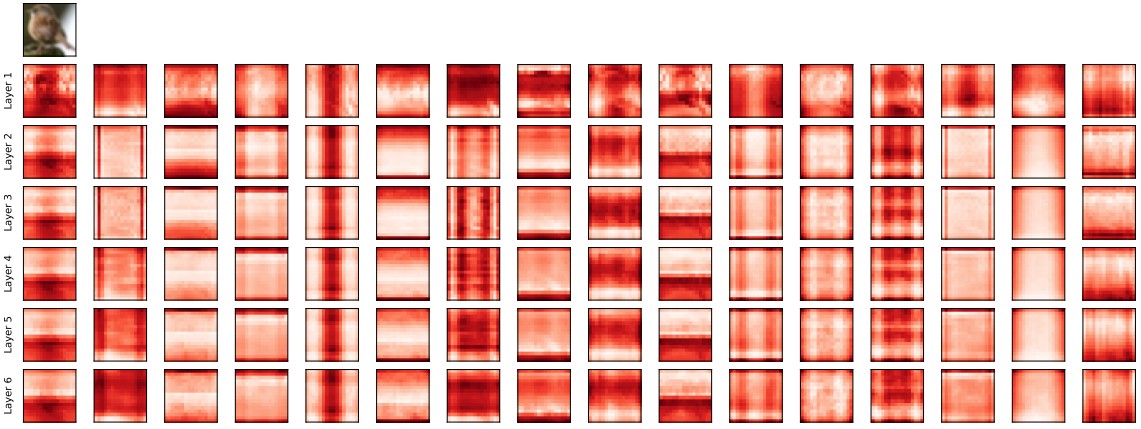

(b) Average Attention visualization

Figure 17: Attention probabilities for a model with 6 layers (rows) and 16 heads (columns) using hierarchical learned embedding w/ content.

## B.7 Hierarchical SAN Pairwise

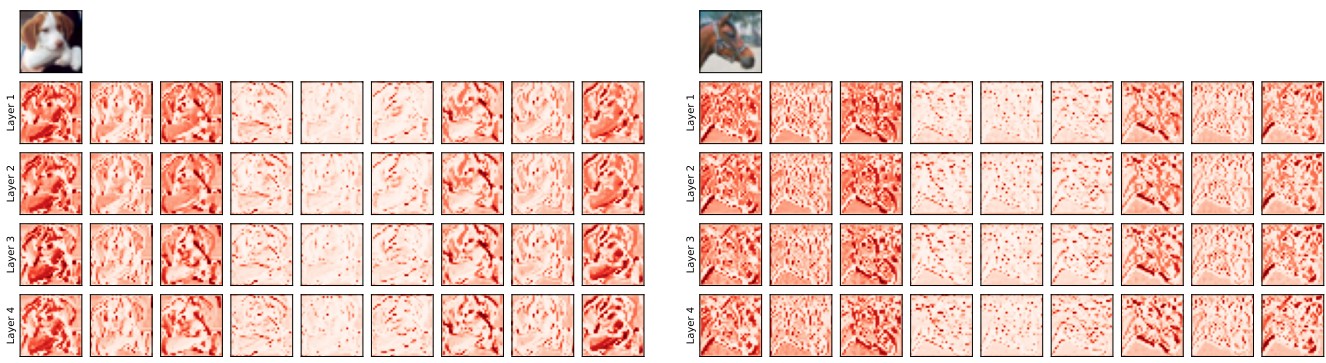

(a) Average Attention visualization

(b) Average Attention visualization

Figure 18: Attention probabilities for a model with 4 layers (rows) and 9 heads (columns) using hierarchical SAN Pairwise.

## B.8 Hierarchical SAN Patchwise

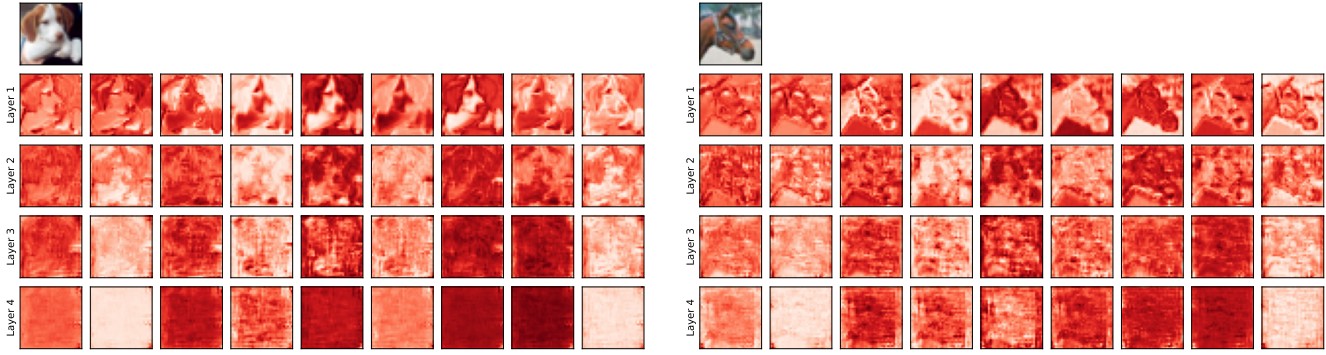

(a) Average Attention visualization

(b) Average Attention visualization

Figure 19: Attention probabilities for a model with 4 layers (rows) and 9 heads (columns) using hierarchical SAN Patchwise.

## B.9 Vision Transformer (VIT)

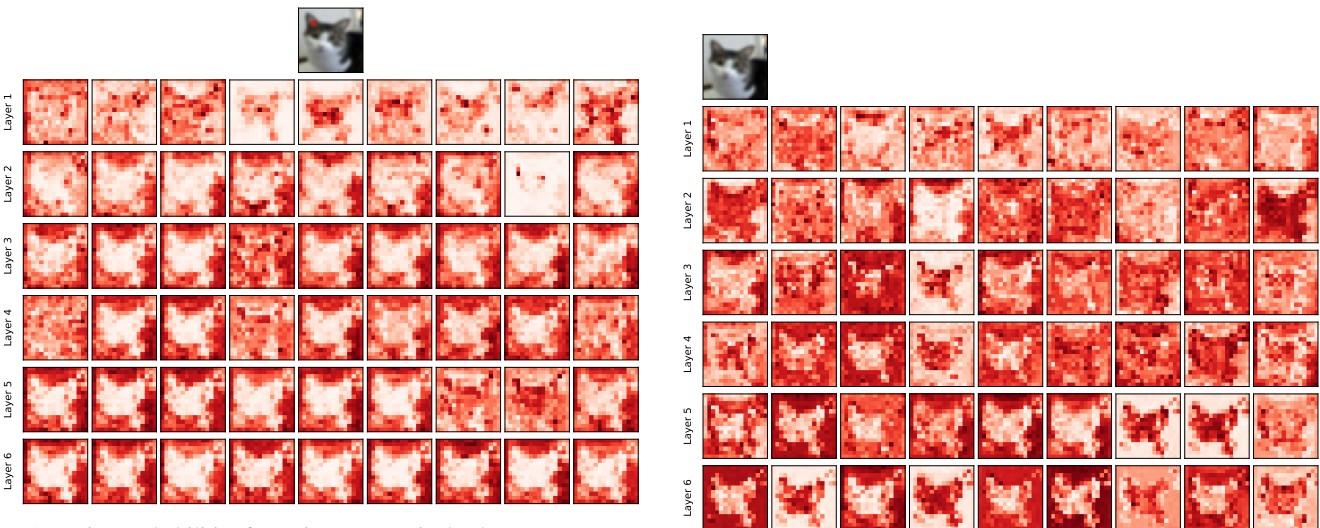

(a) Attention probabilities for a given query pixel. The query (red square) is on the cat's ear

(b) Average Attention visualization

Figure 20: Attention probabilities for a model with 6 layers (rows) and 9 heads (columns) using VIT with patch size $2 \times 2$.

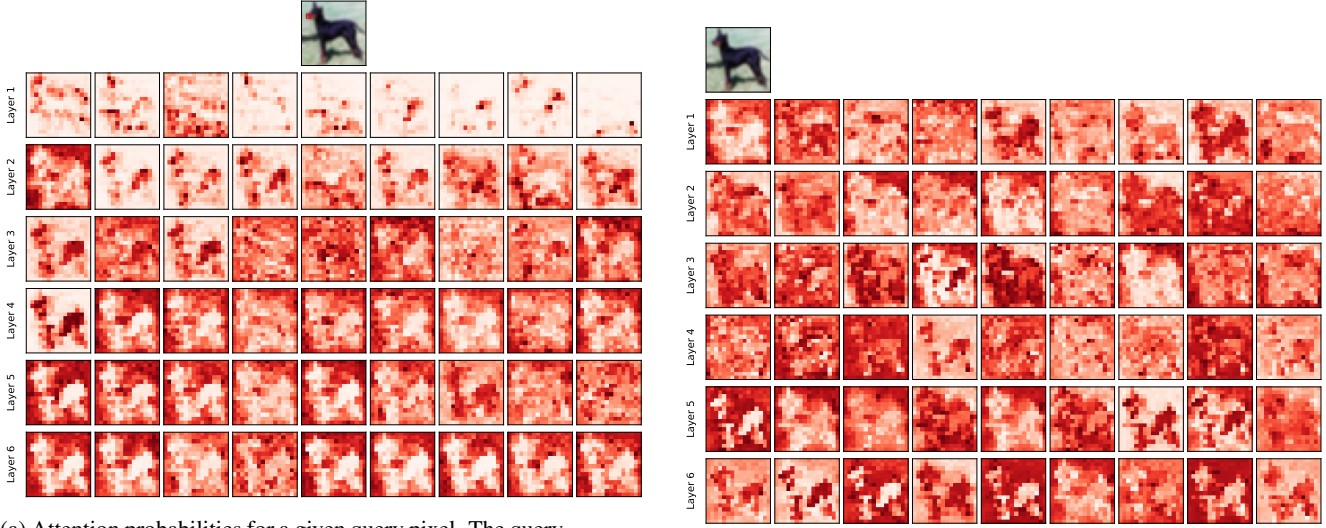

(a) Attention probabilities for a given query pixel. The query (red square) is on the dog's snout

(b) Average Attention visualization

Figure 21: Attention probabilities for a model with 6 layers (rows) and 9 heads (columns) using VIT with patch size $2 \times 2$.

## B.10 Hierarchical Vision Transformer (VIT)

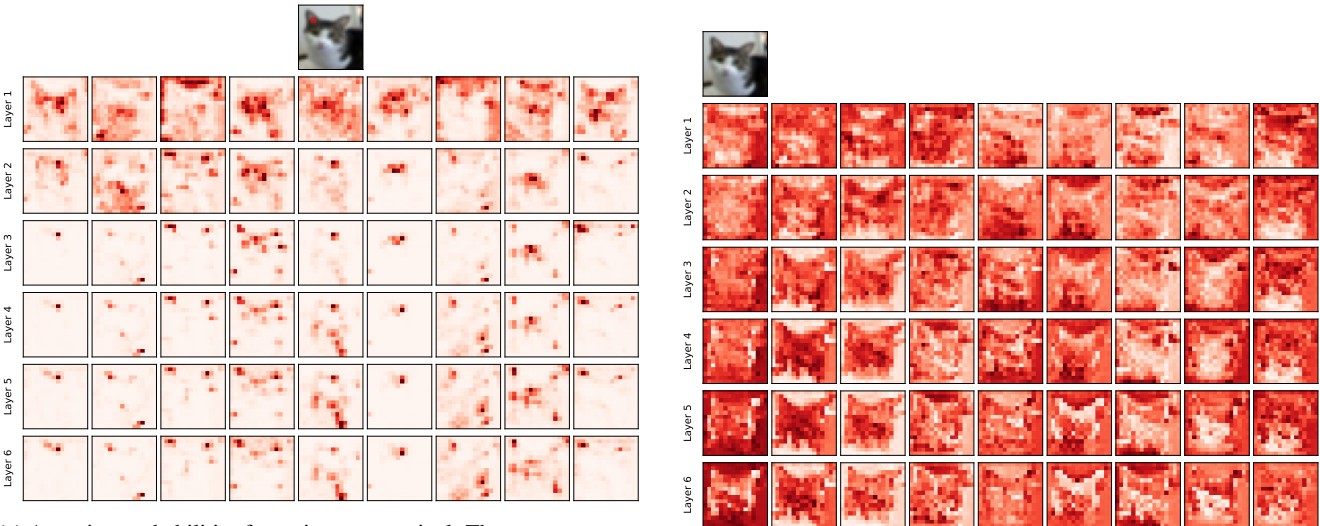

(a) Attention probabilities for a given query pixel. The query pixel (red square) is on the cat's ear.

(b) Average Attention visualization

Figure 22: Attention probabilities for a model with 6 layers (rows) and 9 heads (columns) using hierarchical VIT with patch size $2 \times 2$.

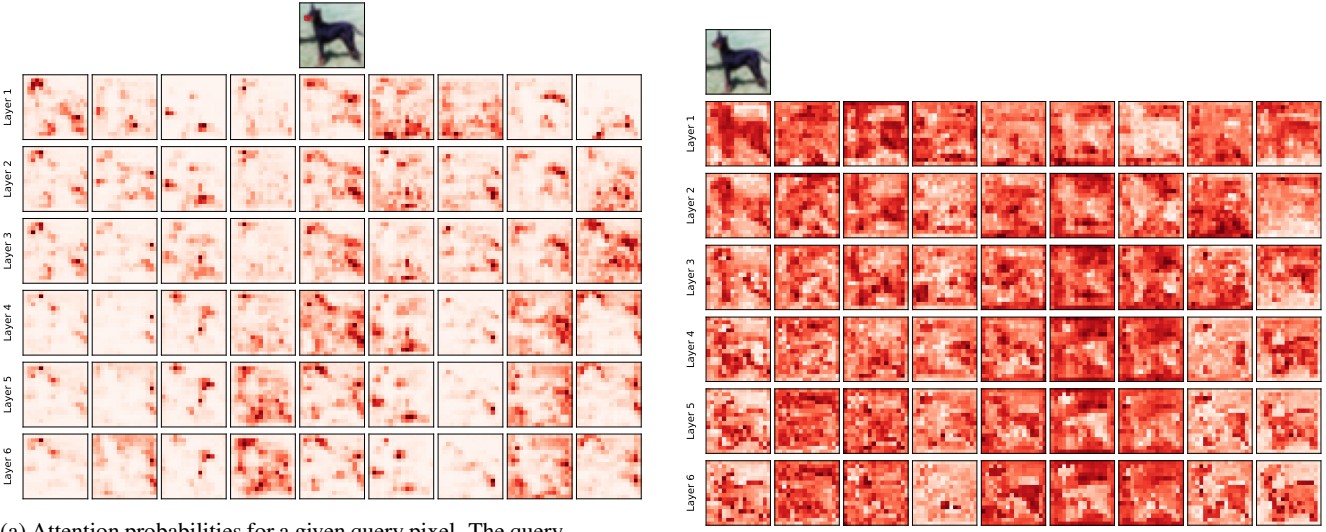

(a) Attention probabilities for a given query pixel. The query pixel (red square) is on the dog's snout.

(b) Average Attention visualization

Figure 23: Attention probabilities for a model with 6 layers (rows) and 16 heads (columns) using hierarchical VIT with patch size $2 \times 2$.

