# OpenReview forum: "[Re]: On the Relationship between Self-Attention and Convolutional Layers"
_ML_Reproducibility_Challenge/2020 — Reject_

### Official Review · AnonReviewer3 · 2021-02-14
**Presented results beyond the scope of the original work, but needs more work on the reproducibility front**

**Rating:** 4
**Confidence:** 3

**Review:**

**Reproducibility summary:** The report does not have the mandatory reproducibility summary.

**Scope of reproducibility:** In Section 3, the authors state that the aim of the section is "to examine whether self-attention layers in practice do actually learn to operate like convolutional layers when trained on standard image classification tasks."

**Code:** It's unclear whether the authors used the original authors' code (which is publicly available) or not. The authors say they "closely follow the official implementation for reproducing the three embedding schemes and attention mechanisms" (line 98) but do not discuss how their implementation is similar to or different from the official implementation. The authors' codebase is clean and organized, but lacks documentation.

**Communication with the original authors**: The authors did not discuss whether they communicated with the original authors.

**Hyperparameter search:** The authors did not conduct a hyperparameter search.

**Ablation study**: The authors did not conduct any ablation studies.

**Discussion on results:** The authors reproduced the original paper's CIFAR-10 experiments and reported similar findings. The authors' discussion of their results is on the shorter side (2 paragraphs in Section 3.3).

**Recommendations for reproducibility:** The authors did not provide recommendations for reproducibility.

**Results beyond the paper**: This report contains several results beyond the original paper. In Section 4, the authors discuss two recent works that attempt to replace convolutions with self-attention. While I found the introduced works interesting, the section lacks discussion of how these works relates to the original paper. In Section 5, the authors propose a new attention operation named Hierarchical Attention (HA) and demonstrate its effectiveness against normal Self-Attention (SA). As the authors emphasize, this operation improves accuracy while using substantially less number of parameters.

**Overall organization and clarity:** Overall, the report was organized and clearly written. However, the authors did not include the mandatory reproducibility summary, and did not discuss their experience reproducing the paper (e.g. methodology, computational requirements, what was easy/difficult, communication with the original authors). While the authors presented results beyond the scope of the original work, the report needs more work on the reproducibility front. I also suggest keeping the citation style consistent and using appropriate labels and references to reference tables and figures.

**Familiar With The Original Paper:**

I have read the original paper

**Reproducibility Summary:**

Summary is missing from the report

---

### Official Review · AnonReviewer1 · 2021-03-01

**Rating:** 3
**Confidence:** 3

**Review:**

***Reproducibility Summary***
Unfortunately the authors have not included the mandatory first page reproducibility summary, and following the reviewer guidelines the submission is liable to desk rejection.

***Scope of reproducibility***
The paper does not state clearly the scope of the reproducibility. The authors reimplement the main experiment in the paper on Cifar, for which they obtain similar results. They also reproduce a similar experiment where they visualize the attention maps, however their attention maps differ significantly from those shown in the paper and there is no in-depth analysis of them.

***Code***
The authors have implemented their own code to reproduce the paper. Unfortunately, their code is attached but is not de-anonymized.

***Communication with original authors***
It is not clearly stated in the paper whether the authors of the reproducibility report contacted the original authors of the paper (or at least I couldn't find this information easily). This information could have been readily available if the authors had included a reproducibility summary.

***Hyperparameter search and ablation study***
The authors conducted limited hyperparameter searches and ablation studies, but the experiments in the original paper are limited in this regard too and therefore it does not seem highly relevant in the evaluation of this report.

***Discussion on results***
The authors do not discuss the results in depth. Most of the conclusions are limited to re-stating the results obtained without properly discussing the differences between their results and the ones found in the original paper. This can be seen in the discussion for the filter results, for example.

***Recommendations for reproducibility***
The authors do not directly provide recommendations to improve the reproducibility of the original paper.

**Results beyond the paper***
The authors put the focus of this report on a new type of attention they propose. Unfortunately, this is at the expense of the quality of the rest of the report, and as part of the reproducibility challenge the paper is does not focus enough on reproducing the original paper.

***Overall clarity***
The article could be better organized, providing first a reproducibility summary.

***Overall rating***
I argue for the rejection of this article based on the omission of the mandatory reproducibility summary, the lack of clarity and the focus on the results beyond the paper but not on reproducing the results of the original paper.

**Familiar With The Original Paper:**

I have read the original paper

**Reproducibility Summary:**

Summary is missing from the report

---

### Official Review · AnonReviewer2 · 2021-03-02
**Good submission with clear insights**

**Rating:** 5
**Confidence:** 4

**Review:**

The authors do not provide reproducibility summary and the manuscript does not match the expected template. There is no discussion about the scope of reproducibility nor any discussion over what was easy/hard. Missing evaluations from the appendix section of the paper.

Overall, authors are able to replicate the experiments reported in the original paper. They also show additional experiments with other choice of attention-based networks which is interesting. Current analysis lacks visualizations for 'centers of attention' which is important. Why is the #params in table 1 and 2 same ? Is the accuracy(paper) in table 2 correct comparison ?

It is commendable that along with reproducing results, the authors propose hierarchical attention operation and evaluate it's performance. They show that it resolves the computational memory requirement as intended. However, keeping the scope of this venue in consideration the paper does not match the expected format and lacks detailed analysis.

**Familiar With The Original Paper:**

I have read the original paper

**Reproducibility Summary:**

Summary is missing from the report

---

### Decision · Program_Chairs · 2021-03-31

**Decision:**

Reject

**Comment:**

This report doesn't fully reproduce the results of the original paper, and lacks the detailed analysis necessary for acceptance